Monitoring deforestation, forest health, and environmental criticality in a protected area periphery using Geospatial Techniques

Withanage Neel Chaminda 1 2
Mishra Prabuddh Kumar prab19@gmail.com prabuddh@shivaji.du.ac.in 3
Abdelrahman Kamal 4
Singh Rajender 3
1 School of Geographical Science, Southwest University , Beibei District , Chongqing, PR China , China
2 Department of Geography, Faculty of Humanities and Social Sciences, University of Ruhuna , Wellamadama, Matara , Sri Lanka
3 Department of Geography, Shivaji College, University of Delhi , New Delhi , Delhi , India
4 Department of Geology and Geophysics, King Saud University , Riyadh , Saudi Arabia
Kent Clement
Electronic publication date: 2024 Jul 18
Publication date: 2024
Volume: 12
Electronic Location ID: e17714
Received 2024 Mar 3; Accepted 2024 Jun 18
Copyright: ©2024 Withanage et al.
Copyright year: 2024
Copyright holder: Withanage et al.
License: This is an open access article distributed under the terms of the Creative Commons Attribution License, which permits unrestricted use, distribution, reproduction and adaptation in any medium and for any purpose provided that it is properly attributed. For attribution, the original author(s), title, publication source (PeerJ) and either DOI or URL of the article must be cited.
License URL: https://creativecommons.org/licenses/by/4.0/

Keywords: Deforestation, Environmental criticality index, Forest health, Land use, NDVI, Vegetation condition index

Funding: Researchers Supporting Project number (RSP2024R351), King Saud University, Riyadh, Saudi Arabia This work was supported by the Researchers Supporting Project number (RSP2024R351), King Saud University, Riyadh, Saudi Arabia (financial support for APC of this article). The funders contributed to the conceptualization, data analysis, preparation of the manuscript. The funders did not have a role in data collection or decision to publish.

==============================
Protected areas in South Asia face significant challenges due to human disturbance and deforestation. The ongoing debate surrounds the recent surge in illegal encroachment of forest buffer zones in the Musali divisional secretariat division (DSD), which has led to a significant loss of forest cover over the past three decades. In this context, detecting changes in forest cover, assessing forest health, and evaluating environmental quality are crucial for sustainable forest management. As such, our efforts focused on assessing forest cover dynamics, forest health, and environmental conditions in the DSD from 1988 to 2022. We employed standardized image processing techniques, utilizing Landsat-5 (TM) and Landsat-8 (OLI) images. However, the forest area in the DSD has shown minimal changes, and environmental conditions and forest health have illustrated considerable spatial-temporal variations over the 34 years. The results indicated that 8.5 km2 (1.9%) of forest cover in the DSD has been converted to other land use classes. Overall, the Normalized Difference Vegetation Index (NDVI) has declined over time, while Land Surface Temperature (LST) exhibits an increasing trend. The regression results demonstrated a robust inverse relationship between LST and NDVI. The declining vegetation conditions and the increasing LST contribute to an increase in environmental criticality. The derived maps and indices will be beneficial for forest authorities in identifying highly sensitive locations. Additionally, they could enable land use planners to develop sustainable land management strategies.

Introduction

Changes in land use/land cover (LULC) reflect the impacts of socioeconomic factors on the environment and human–environment interactions (Kayet & Pathak, 2015; Wijesinghe & Withanage, 2021; Withanage, Mishra & Jayasinghe, 2024). Thus, updated and reliable land information is crucial for both current and future land use planning (Wijesinghe & Withanage, 2021). Employing multi-temporal satellite imagery for LULC change detection helps in comprehending landscape dynamics (Rawat & Kumar, 2015). Previous studies have indicated that LULC changes have resulted in significant damage to forest cover, leading to a rapid pace of deforestation. The growing population, urbanization, and infrastructure development have contributed to an imbalance between the supply and demand for forest products, posing a threat to forest ecosystems (Koellner et al., 2008). Deforestation has led to the disappearance of 420 million hectares of forest between 1990 and 2020 (FAO, 2022 and FAO, 2014). The recent emphasis on non-consumptive use of forests has garnered the attention of forest conservation authorities worldwide, leading to sustainable forest management (Ranagalage et al., 2020). Reliable quantification of forest cover and vegetation health is essential for developing comprehensive forest resource management guidelines.

Changes in forest cover play a significant role in influencing LST, which in turn can directly impact energy balance, evapotranspiration, and precipitation patterns, ultimately altering vegetation conditions (Jaafar et al., 2020; Deng et al., 2018; Culf et al., 1996). Given the challenges associated with in-situ observations, sensor platforms can be employed to retrieve LST effectively (Jaafar et al., 2020). Numerous studies have utilized Landsat data to analyze forest cover dynamics and health, often by analyzing NDVI (Jaafar et al., 2020; Culf et al., 1996; Deng et al., 2018; Malik, Shukla & Mishra, 2019; Jenerette et al., 2006). The NDVI is a basic index used to assess vegetation cover, vegetation health, and photosynthetic activity and some studies implied that NDVI is an indicator of the link between LST and vegetation conditions (Jaafar et al., 2020; Anbazhagan & Paramasivam, 2016; Peng et al., 2014; Yuan et al., 2017). Additionally, it can be used as a proxy for biomass accumulation (Jaafar et al., 2020). NDVI is used to create other indices, such as the Vegetation Condition Index (VCI), by normalizing long-term satellite-based NDVI data (Yin et al., 2024). Moreover, LST and NDVI serve as indicators of changing environmental quality. The Environmental Criticality Index (ECI) is calculated based on the ratio between LST and NDVI by normalizing each layer. Accordingly, previous researchers have employed the ECI to assess varying levels of risk associated with environmental conditions (Senanayake, Welivitiya & Nadeeka, 2013; Ranagalage, Estoque & Murayama, 2017; Saputra, Jamadi & Sari, 2023).

While there have been numerous studies quantifying forest cover using remote sensing data in other areas (Wickramagamage, 1998; Fernando & Edirisuriya, 2016; Ranagalage et al., 2020), it is challenging to find similar studies in our study area. Therefore, our study fills this gap by measuring forest cover dynamics and evaluating forest health and environmental conditions between 1988 and 2022 in Musali DSD which located in the northern periphery of Vilpattu National Park, Sri Lanka. The study objective was to assess changes in forest cover, vegetation health, and environmental conditions using the Landsat time series dataset. Three standardized indices: LST; NDVI; and VCI, were employed to evaluate changes in forest health. Additionally, environmental condition was analyzed using the ECI.

Materials and Methods

Study area

Musali is one of the five DSD located in the Mannar district of northwestern Sri Lanka. This DSD located within latitudes 8°31′26″N, 8°49′2″N and longitudes 8°3′2″E, 79°56′29″E. Nanaddan DSD bounds it to the North, while to the South lie the Anauradhapura and Puttalam districts. The Madu DSD marks its Eastern border, with the Gulf of Mannar stretching along its Western edge (Fig. 1). The total extent of the DSD is 475 Km2 with 20 Grama Niladhari Divisions (GNDs). The population of the DSD is comparatively low and there was 29,011 population in 2021 with showing density is as 61 persons per km2 (Department of Census and Statistics, 2022). The primary livelihood activities in the DSD include agriculture, fishing, and animal husbandry.

Figure 1 Location: (A) Sri Lanka; (B) DSD in Mannar district; (C) Landsat 4,3,2 composite of Musali DSD.

Maps were created by authors using United States Geological Survey Earth Explorer Landsat images (https://earthexplorer.usgs.gov/) and Sri Lanka Survey department hard copy maps of DSD and Sri Lankan boundary.

While the annual and monthly average rainfall for Sri Lanka stands at 2,397 mm and 200 mm respectively, the Musali experiences an annual average rainfall of 960 mm, with monthly averages ranging from 74 mm to 134 mm. Thus, DSD is characterized as one of the driest regions in Sri Lanka, with evapotranspiration reaching 2,135 mm/year, placing it within a semi-arid climate zone (Athauda et al., 2024). The average humidity in the DSD is approximately 75%, attributed to its location within the semi-arid zone. The DSD typically encountering only the tail end of the northeastern monsoon. In April, aside from the monsoon, the DSD also experiences rainfall from conventional rains. For the remainder of the year, DSD experiences a long dry season (Land Use Policy Planning Department, 2016). The rainfall pattern is very different from the southwestern side of Sri Lanka. The annual minimum temperature in DSD is 26 °C while annual maximum temperature is 35 °C (Department of Census and Statistics, 2022). There is little variation in the temperature during the entire year. The dry season is possibly the longest in Sri Lanka, and a brief and light monsoon season during the winter months. Agro-ecologically, the DSD falls into the low-country dry zone-DL3 (Land Use Policy Planning Department, 2016).

Most areas in DSD is comprised of lowlands (elevation: 0 to 12 m) with some scattered undulating surface topography (slope: 0° to 5°). The predominant geological formation comprises Miocene limestone and Quaternary deposits. The soil types in the area encompass yellow-brown sands, dune and beach sands, as well as lagoonal deposits (Athauda et al., 2024). Along the coastal area, sandy soils (Regosols) predominate, while saline and marshy lands are prevalent in the low-lying areas (Land Use Policy Planning Department, 2016). Various forest and vegetation types can be identified in the DSD, encompassing dry monsoon forest, riverine dry forest, mangroves, and dense forest.

Materials

The study primarily used Landsat-5 Thematic Mapper (TM) and Landsat-8 Operational Land Imager (OLI) multispectral image series to investigate the spatial and temporal variations in vegetation cover, vegetation health, and environmental conditions (United State Geological survey, 2023). All images were taken during the dry period of the southwest monsoon (February and August) with less than 5% cloud cover, sourced from the USGS Earth Explorer (https://earthexplorer.usgs.gov/) for the year 1988, 1996, 2009 and 2022. Table S1 summarizes the Landsat data specifications used for deriving LST, NDVI, ECI, and for mapping forest cover changes. To create the study area boundary map and the Sri Lanka boundary map, hard copy maps available for purchase from the Sri Lanka Survey Department were used. ArcMap 10.8 software (ESRI, February, 2020) was used for remote sensing image processing, while regression analysis was performed using MS Office Excel (2013) spreadsheet program.

Figure 2 Methodological flowchart of the study.

Maps were created by authors using United States Geological Survey Earth Explorer Landsat images (https://earthexplorer.usgs.gov/) and Sri Lanka Survey department hard copy maps of DSD and Sri Lankan boundary.

Methods

Several steps were followed to create forest cover maps and other indices, including image pre processing, classification, accuracy assessment, change detection, and the calculation of NDVI, LST, VCI, and ECI. Finally, regression analysis was performed to find the correlation between NDVI and LST for the concerned period. The methodical flowchart is represented in Fig. 2.

Image pre processing

Pre processing of data involved several key steps to ensure that the data is clean, accurate, and ready for analysis. Firstly, radiometric calibration was performed by converting raw digital numbers (DN) to radiance or reflectance values. This process corrected sensor-specific biases. Secondly, we removed the effects of the atmosphere on the reflectance values by employing Dark Object Subtraction (DOS). To maintain spatial consistency, we performed geometric correction by projecting all data layers into the WGS 1984, UTM zone 44N projection system. In the image subsetting phase, we clipped the images to the areas of interest (AOI) using the DSD boundary. Then, we adjusted the spatial resolution of the Landsat images using the nearest neighbor resampling technique while maintaining a 30 m spatial resolution.

Image classification

The features in the images were retrieved to predetermine the forest cover changes in Musali DSD. During classification, five land use classes were identified: forest, agriculture, water bodies, built-up/settlements, and others, as shown in Table S2. There are various statistically driven supervised classification algorithms, such as maximum likelihood, parallelepiped, and Mahalanobis. The chosen classification method for this study is maximum likelihood classification (MLC). MLC, a widely used algorithm for LULC classification, relies on statistical sampling using probability density functions to detect predefined sets of LULC classes (Alawamy et al., 2020). Moreover, this method is widely used because fine-resolution satellite remote sensing data are comparatively inexpensive sources for LULC mapping (Weng, 2002; Dissanayake et al., 2019). It relies upon the likelihood that a pixel belongs to a specific class. One well-known method, MLC, based on the Bayesian equation, calculates the likelihood D of unknown measurement vector X, belongs to Eq. (1) (Mohajane et al., 2018): (1) D=Inac−0.5InCovc−0.5X−McTCovc−1XMc

In the equation the weighted distance represent by D; particular class is c; the measurement vector of the candidate pixel is represented by X. The mean vector of the sample of class c represented by Mc. The percent probability of the candidate pixel being a member of class c represented by ac. The covariance matrix of the pixels in the sample of class c is presented by Covc. Covc is the determinant of Covc (matrix algebra). Covc − 1 is is the inverse of Covc (matrix algebra). ln = natural logarithm function; and T = transposition function (matrix algebra) (Mohajane et al., 2018).

The majority filtering method has been employed to rectify the errors that may arise during the classification process. This approach has been utilized by previous researchers as well to bolster the reliability of their results (Weng, 2002; Dissanayake et al., 2019). Figure S1 illustrates the LULC information provided in Table S2. These images were collected from the Google Earth Pro. August 25, 2022, was the date for selecting the images as it is the closest available date to offer an illustrative representation of the LULC information.

Accuracy assessment

Securing the quality of classified images is paramount in every LULC change detection process. To achieve this, a stratified random sampling technique was employed in the procedure, ensuring coverage of all LULC classes, with 600 Ground Control Points (GCP) created each year (Wu & Murray, 2003). Subsequently, Google Earth Pro historical imagery served as reference data for accuracy assessment. Four widely used accuracy metrics were computed: Overall Accuracy (OA), Producer’s Accuracy (PA), User’s Accuracy (UA), and Kappa Coefficients (K).

OA represents the overall percentage of correctly classified LULC classes, calculated by dividing the number of accurately classified land cover pixels by the total number of pixels in the datasets using Eq. (2) (Yuh et al., 2023; Olofsson et al., 2014; Weng, 2002; Dissanayake et al., 2019; Estoque & Murayama, 2017). (2) OA=1N∑ii=1nPii

where, OA is overall accuracy; N is total samples number; n is total categories number; and Pii is correct classifications number of ith sample in confusion matrix.

PA measures the percentage accuracy of individual LULC classes within a map, determined by dividing the number of correctly classified pixels in a specific land cover class by the total number of pixels belonging to that class in the reference data. Misclassified pixels within this metric are known as errors of omission and was calculated using as Eq. (3): (3) PA=Correctly classified number pixel in each categoryCorrectly classified total number pixels in that category (column total)

UA evaluates the reliability of a given land cover map concerning its agreement with ground observations. It is calculated by dividing the number of correctly classified pixels in a specific land cover class by the total number of pixels classified within that class. Similar to Producer’s Accuracy, misclassified pixels in this metric are referred to as errors of omission. Equation (4) is as below: (4) UA=Correctly classified number pixel in each categoryCorrectly classified total number pixels in that category(row total)

K indicates the level of agreement between test and validation data in a generated land cover map. It is based on the probability of the test data closely matching the validation data during the land cover mapping process and is highly correlated with overall accuracy. K coefficient was utilized as a metric to gauge the agreement between classified results and actual conditions, thereby determining the levels of accuracy. A threshold of 0.75 or higher was set, indicating sufficient agreement for actionable insights based on the image. The Eq. (5) was used to calculate the K (Alqurashi & Kumar, 2014). The results of the accuracy assessments are presented in Table S3. (5) K=N∑i=jkxii− ∑i=1kxi+×x+iN2− ∑i=1kxi+×x+i

where k is the number of rows in the matrix; xii is the number of observations in row i and column i; xi+ and x+i are the marginal totals of row k and column i. N is the number of observations (Alqurashi & Kumar, 2014).

Change detection

A change detection method was used to cross-tabulate LULC data between different periods after image classification: (i) 1988 vs. 1996, (ii) 1996 vs. 2009, (iii) 2009 vs. 2022, and (iv) 1988 vs. 2022.

Retrieval of LST

LST is commonly defined as the temperature at the interface between the Earth’s surface and its atmosphere and this serves as a critical indicator in all physical processes concerning surface energy and water balance at both micro and macro spatial scale (Malik, Shukla & Mishra, 2019; Sobrino, Jiménez-Muñoz & Paolini, 2004). This plays a pivotal role in surface processes and boasts a wide array of applications, including vegetation stress monitoring, climate change studies, environmental assessment, and urban climate analysis. With the availability of large-scale remote sensing data, near-surface air temperature measurements can be effectively monitored and utilized to retrieve the temperature of various LULC surfaces (Malik, Shukla & Mishra, 2019). In our study, Landsat 5 and Landsat 8 data were employed to analyze LST and investigate the impact of forest degradation. The LST was obtained by Thermal Infrared (TIR) band 6 of Landsat 5 and TIR band 10 and 11 of Landsat 8 by converting DN values into radiance values (Jaafar et al., 2020; Ranagalage, Estoque & Murayama, 2017; Dissanayake et al., 2019). Here we used thremal bands holding brightness temperatures which are represented in Kelvin. Before retrieval, the LST land surface emissivity values was derived using Eq. (6): (6) ɛ=mPV+n

where ɛ represents land surface emissivity; m represents (ɛv − ɛs) − (1 − ɛs) ɛv; Pv represents the amount of vegetation; n represents ɛs + (1 − ɛs) ɛv ; ɛs is soil emissivity; ɛv is the vegetation emissivity; and F is a shape factor (Ranagalage, Estoque & Murayama, 2017; Dissanayake et al., 2019). Here we used m = 0.004 and n = 0.986 as in previous research (Jaafar et al., 2020; Ranagalage, Estoque & Murayama, 2017; Dissanayake et al., 2019). The proportion of vegetation (PV) is derived from Eq. (7). (7) PV=NDVI−NDVImin/NDVImax−NDVImin2

where NDVI is the normalized difference vegetation index derived from Eq. (9) as in sub section below. The NDVImin and NDVImax are the minimum and maximum values of the NDVI, respectively. Then emissivity corrected LST were retrieved using Eq. (8). (8) LST°C=TB1+λ×TB/pInɛ

where TB = Landsat TM Band 6 at-satellite brightness temperature; λ = wavelength of emitted radiance (λ = 11.5 µm for Landsat TM Band 6, λ = 10.8 µm for Landsat TIRS Band 10) (Ranagalage, Estoque & Murayama, 2017; Dissanayake et al., 2019); p = h × c/σ(1. 438 × 10 –2 mK), σ = Boltzmann constant (1. 38 × 10 − 23 J/K), h = Planck’s constant (6. 626 × 10 − 34 Js),c = velocity of light (2. 998 × 108 m/s), ɛ is the land surface emissivity. Last, the LST values of Kelvin were converted into degrees Celsius (°C).

Derivation of NDVI

The results of spectral analysis are typically summarized into vegetation indices, which establish relationships between reflectance across two or more wavelength intervals or bands in satellite images. Measurement of such indices serves as an especially valuable tool for precision agriculture and vegetation analysis (Fernández-Alonso, Hernández & Torres-Costa, 2023). Various sensors and multi spectral devices are extensively utilized nowadays to assess the condition of vegetation, including chlorophyll meters, canopy reflectance sensors, and Plant-O-Meter (Padilla et al., 2018; Kitić et al., 2019). NDVI is the most common vegetation index, frequently employed as an indicator of chlorophyll content and overall vegetation health (Fernández-Alonso, Hernández & Torres-Costa, 2023). The NDVI is a vital remote sensing metric, acting as a key indicator for assessing vegetation coverage and growth. This quantifies vegetation through the spectral reflectance and derived from the ratio of the difference to the sum of values in the red and near-infrared bands, with values typically ranging from −1.0 to 1.0 (Yin et al., 2024). Near infrared (NIR) (band 4 in TM and band 5 in OLI) and red bands (band 3 in TM and band 4 in OLI) of Landsat are needed to retrieve the NDVI. Vegetation density and chlorophyll activity variations were measured using the Landsat images which were taken during the dry period of the study area using Eq. (9) (Jaafar et al., 2020; Ranagalage, Estoque & Murayama, 2017; Dissanayake et al., 2019). (9) NDVI=NIR−RED/NIR+RED

The retrieved maps were classified upon their values.We used the thresholding method to classify NDVI into four categories: non vegetation (less than 0) low-density vegetation (0–0.2), moderate-density vegetation (values between 0.2 and 0.5), and high-density vegetation (values above 0.5) following the similar work conducted by Mohajane et al. (2018).

Derivation of VCI

VCI is a widely used drought index which developed by Kogan (1990) for monitoring vegetation drought stress and estimating drought trends. Kogan (1990) developed this index to enhance the weather-related components in the NDVI value (Ha, Uereyen & Kuenzer, 2023). It is constructed by normalizing long-term satellite-based NDVI data. The VCI offers several advantages, such as eliminating envelope signal variations in NDVI and accounting for regional climate diversity. It is also easy to construct using only NDVI data. Consequently, it has various applications, including drought investigation, crop yield estimation, and vegetation dynamics assessment (Yin et al., 2024; Ha, Uereyen & Kuenzer, 2023). This helps to compare current NDVI values with past year NDVI values of the same season. Equation (10) was used to calculate VCI for the selected years (Yin et al., 2024; Ha, Uereyen & Kuenzer, 2023; Kogan, 1990; Dutta et al., 2015). (10) VCI=NDVI−NDVImin/NDVImax−NDVImin∗100

where NDVI indicates the value of a specific pixel in that particular month, NDVImax and NDVImin show the multi-layer highest and lowest NDVI values in the same period. The VCI values represented as % and 0 is for extreme drought conditions and 100 is optimal vegetation health. Following (Ha, Uereyen & Kuenzer, 2023; Kogan, 1990; Dutta et al., 2015), we classified VCI values into five categories upon drought severity: non drought (100%–50%), mild drought (50%–30%), moderate drought (30%–20%), severe drought (20%–10%), and extreme drought (less than 10%).

Derivation of ECI

Due to rising LST and declining NDVI, the environment is in a critical state, as measured by the ECI. Increases in LST have been shown to directly correlate with ECI, while decreases in NDVI have been reported to have an inverse correlation with ECI (Senanayake, Welivitiya & Nadeeka, 2013; Saputra, Jamadi & Sari, 2023). Based on the ratio between LST and NDVI the ECI is calculated to identify environmentally critical areas (Senanayake, Welivitiya & Nadeeka, 2013; Ranagalage, Estoque & Murayama, 2017)). The LST and NDVI layers are used to derive the ECI using Eq. (11). The retrieved NDVI and LST layers were first normalized using the histogram equalization method, which ranges pixel values between 1–255 (Ranagalage, Estoque & Murayama, 2017). The higher the ECI value, the more environmentally critical. The spatial variation of ECI over the study area was interpreted as high, medium, and low. (11) ECI=LSTStretched1−255/NDVIStretched1−255.

Results

Forest cover change in Musali DS

Figures 3 and 4 illustrate the spatial and temporal variations of the LULC in the study area. According to the findings presented in Table 1, the forest area has decreased from 348.7 km2 (73.5%) to 340 km2 (71.6%) over 34 years. The urban areas and settlements have expanded from 13.9 km2 (2.9%) to 28 km2 (5.8%) between 1988 and 2022, primarily driven by population growth and other socioeconomic factors. Agricultural lands have decreased by 2.2 km2 over this period. The area of water bodies has diminished by approximately 2.4 km2, while other land uses (sand and salt marshes) have decreased by approximately 1 km2. Between 1988 and 1996, forest cover experienced a significant decrease, with a negative change of approximately −9.7 km2 (Table 2). Consequently, the net change in forest area over the 34-year period is −8.09 km2, reflecting a gain of 20.5 km2 and a loss of 28.6 km2. The primary cause of the decline in forest area is the expansion of agriculture and settlements, as depicted in Fig. S2.

Figure 3 Land use in Musali DSD: 1988; 1996; 2009; 2022.

Maps were created by authors using United States Geological Survey Earth Explorer Landsat images (https://earthexplorer.usgs.gov/) and Sri Lanka Survey department hard copy maps of DSD and Sri Lankan boundary.

Figure 4 Land use statistics in Musali DSD: (A) 1988; (B) 1996; (C) 2009; (D) 2022.

Within the specified time frame, approximately 4.8 km2 of forest land was converted into agricultural areas, specifically for paddy fields (Fig. S2). Additionally, a total of 2.75 km2 of forest land was converted into built-up areas. Between 1988 and 1996, the majority of the scattered forest patches in the northwestern area (Fig. 5A) were converted into agricultural areas. Between 1996 and 2009, a significant portion of the forest lands adjacent to roadways, as seen in Figs. 5C and 5D, were taken over by human settlements and developed areas. Over time, a significant portion of agricultural land owned by Kondachchi Plantation Limited within Vilpattu National Park has been converted into forest areas (Fig. 5E). Between 2009 and 2022, forest areas along the northern boundary, southeastern, and eastern portions experienced conversion into agricultural and residential areas. Over 34 years, there has been a 3% expansion in built-up and settlement areas, totaling approximately 14.35 km2. Additionally, there has been a reduction of forest cover by about 8.09 km2.

Spatial-temporal changes of NDVI and LST

The spatial and temporal distribution of NDVI shows significant variability. Despite deforestation experiencing relatively minor changes over 34 years, the NDVI indicates a significant decline, suggesting a deterioration in the health of the vegetation in the region. The NDVI shows a significant reduction in high-density vegetation areas, which have transitioned into moderate- and low-density vegetation areas over the past 34 years. Additionally, there has been an increase in non-vegetation areas (Figs. 6A–6D). The corresponding descriptive statistics, as shown in Table S4, also support this phenomenon. In 1988, NDVI values ranged from −0.55 to 0.75, indicating extensive high-density vegetation coverage across the study area (Fig. 6A). However, in 2022, this range decreased from −0.18 to 0.54. At all four time points, areas with high NDVI values were found in high-density and moderate-density vegetation areas, while areas with low NDVI values were concentrated in non vegetation and settlement areas, as depicted in Fig. 6B. Between 1988 and 1996, the NDVI indicated a significant decrease in high-density vegetation in the study area while non-vegetation areas expanded (Figs. 6C, 6D). This trend continued further in 2009 and 2022. The areas of Silavatura, Alakaddu, and Putukulam, which have recently been developed and populated, have exhibited a decrease in NDVI values by the year 2022. The northern portion of the region features paddy fields with low NDVI values.

Table 1 LULC changes summary in Musali DSD,1988-2022.

LULC	1988	1996	2009	2022	
	Area (km2)	%	Area (km2)	%	Area (km2)	%	Area (km2)	%	
FC	348.72	73.5	345.56	72.4	342.05	72.01	340.2	71.6	
AG	89.27	18.7	93.54	19.6	84.72	17.8	87.4	18.4	
BS	13.95	2.9	18.28	3.8	26.39	5.5	28.3	5.9	
WB	9.45	1.98	6.57	1.3	4.53	0.95	7.1	1.4	
OT	13.05	2.74	11.05	2.3	16.74	3.5	12.0	2.5	
Total	474.44	99.8	474.44	99.8	474.44	99.8	474.44	99.8	
Notes.

FC Forest cover

AG Agriculture

BS Built-up/settlements

WB Water bodies

OT Others

Table 2 Statistics of forest cover changes in Musali DSD.

	1988–1996	1996–2009	2009–2022	1988–2022	
Forest gain (km2)	14	11.7	6.3	20.5	
Forest loss (km2)	23.7	28.2	7.8	28.6	
Net Change (+/-)	−9.7	−16.5	−1.5	−8.09	

The spatial and temporal changes in LST over 34 years are depicted in Figs. 7A–7D. In 1988, the lowest recorded LST was 22.3 °C, while the highest recorded LST was 32.8 °C. The average LST for that year was 25.2 °C. Nevertheless, the temperature has risen by approximately 6 °C by 2022, reaching a minimum of 24.5 °C and a maximum of 38.8 °C, with an average of 26 °C (Table S5). Areas that are not covered by forests exhibit elevated LST, particularly in built-up and residential areas. Specifically, in the year 2022, new residential areas and deforested areas in Marichchakattu, Kondachchi plantation, and Alakaddu are indicated as prominent hot spots in Fig. 7D. Nevertheless, the areas with high LST in the northern part experienced a significant decrease in LST by 2022 compared to the years 1988, 1996, and 2009. However, the LST in the areas that were deforested between 1988 and 1996 in the northern boundary showed a rise after 2009.

Figure 5 Forest cover change in Musali DSD: 1988; 1996; 2009; 2022.

Maps were created by authors using United States Geological Survey Earth Explorer Landsat images (https://earthexplorer.usgs.gov/) and Sri Lanka Survey department hard copy maps of DSD and Sri Lankan boundary.

Figure 6 Spatial changes of NDVI in Musali DSD: (A) 1988; (B) 1996; (C) 2009; (D) 2022.

Maps were created by authors using United States Geological Survey Earth Explorer Landsat images (https://earthexplorer.usgs.gov/) and Sri Lanka Survey department hard copy maps of DSD and Sri Lankan boundary.

Figure 7 Spatial changes of LST in Musali DSD: (A) 1988; (B) 1996; (C) 2009; (D) 2022.

Maps were created by authors using United States Geological Survey Earth Explorer Landsat images (https://earthexplorer.usgs.gov/) and Sri Lanka Survey department hard copy maps of DSD and Sri Lankan boundary.

Figure 8 illustrates the regression results for the correlation between NDVI and LST for the years 1988, 1996, 2009, and 2022. The scatter plots demonstrate a strong inverse relationship between LST and NDVI across three time points. The coefficient of determination (R2) values for the years 1988, 1996, and 2009 are indeed quite high. Furthermore, there was a noticeable upward trend in the data, indicating the strong predictive ability of NDVI in explaining the spatial changes in LST throughout the three time points, except for 2022 that shows the low R2 value as 0.339.

Figure 8 Regression results between NDVI and LST in Musali DSD: (A) 1988; (B) 1996; (C) 2009; (D) 2022.

Variation of VCI 1988–2022

Examination of the VCI values reveals notable spatial and temporal variations in vegetation health within the DSD across four periods. In 1988, the average VCI value stood at 93.4, yet it plummeted to a mean of 15.2 by 2022. Over the span of 34 years, the standard deviation has escalated from 17.7 to 31.8. Figure 9 depicts the spatial distribution trends of vegetation health at four distinct period. In 1988, vegetation health was good, as most areas of the DSD experienced non-drought or mild drought conditions (Fig. 9A). However, by 1996, vegetation stress had increased significantly due to extreme and severe drought conditions, particularly impacting the northwestern region (Fig. 9B). Although extreme and severe drought conditions were not as prevalent as in 1996, the areas experiencing non-drought and mild drought conditions had further decreased by 2009 (Fig. 9C). In 2022, the vegetation health shows an improvement compared to 2009, with non-drought vegetation coverage expanding once more, except in areas of extreme and severe drought in the northwestern region (Fig. 9D). This phenomenon can be ascribed to the favorable climatic conditions prevalent in year 2022. By 2022, there was a notable increase in vegetation stress observed in the northwestern and southern boundaries compared to the levels recorded in 2009.

Figure 9 Vegetation Condition Index in DSD: (A) 1988; (B) 1996; (C) 2009; (D) 2022.

Maps were created by authors using United States Geological Survey Earth Explorer Landsat images (https://earthexplorer.usgs.gov/) and Sri Lanka Survey department hard copy maps of DSD and Sri Lankan boundary.

ECI in 1988, 1996, 2009, and 2022

Figures 10A–10D displays the ECI of the DSD for the years 1988, 1996, 2009, and 2022. The analysis reveals a steady increase in the ECI value by 2022. Initially recorded at 7.74 in 1988, the ECI has progressively climbed to 9.3, 12.8, and 15.5 in 1996, 2009, and 2022, respectively. According to the distribution maps, the areas with the highest ECI values were primarily situated within the settlements and built-up areas in the DS across all four time points. Generally, dense forested regions exhibiting low ECI values suggest minimal impact on the ecosystem. However,built-up areas along the northwestern boundary, such as Arippu, consistently displayed a notable concentration of ECI values across all four time points, including 1988 (Fig. 9A). Moreover, new concentrations of ECI have emerged in recently deforested regions such as Marichchakattu, Alakaddu, and the Kondachchi plantation area, as illustrated in Fig. 9D.

Figure 10 Environmental Criticality Index in DSD: (A) 1988; (B) 1996; (C) 2009; (D) 2022.

Maps were created by authors using United States Geological Survey Earth Explorer Landsat images (https://earthexplorer.usgs.gov/) and Sri Lanka Survey department hard copy maps of DSD and Sri Lankan boundary.

Discussion

The trends and Drivers of forest cover change

To sum up, our research indicates that over the course of 34 years, there has been a steady rate of deforestation occurring in an area adjacent to the northern boundary of the Vilpattu National Park. This deforestation appears to stem from the establishment and expansion of human settlements in the vicinity. Ranagalage et al. (2020) also found that the deforestation rate in dry zone regions from 2009 to 2020 stood at 3.4%. The gradual pace of deforestation observed in the examined area could have influenced the dynamics of the civil conflict spanning from 1988 to 2009, potentially leading to fewer new settlements and limited expansion of agricultural lands. Furthermore, they found that the civil war resulted in swift forest regeneration across the northern and eastern province. This resurgence can be attributed to the abandonment of human settlements and other disruptions caused by Chena cultivation. Due to the civil war, a considerable portion of the population moved from the study area to different districts within the country. The main factors driving the decline in forest cover in the study area after 2009 were resettlement efforts following the war, infrastructure development, and various rural development projects (Fernando et al., 2015; Fernando & Edirisuriya, 2016; Wickramagamage, 1998; Forest Department Government of Sri Lanka, 2009). Rathnayake, Jones & Soto-Berelov (2020) also observed that LULC changes have affected protected areas in Sri Lanka. They noted that many protected areas situated near district capitals have experienced significant impacts due to rising human population pressure and urbanization. Following 2009, a notable portion of the forested areas in the dry zone underwent conversion into settlements and agricultural lands. This transformation occurred as a result of the majority of displaced individuals returning to their native residences, leading to encroachments on the forest buffer zone within the study area (Fig. S3). Since 2009, certain locations within the study area, such as Karadikkuli, Kondachchi, and Silawatura, situated in close proximity to the Vilpattu forest buffer zone, have experienced encroachment by resettled individuals who have established residences and cultivated agricultural land.

Relationship between changing LST, NDVI, and ECI

While the deforestation rate hasn’t been alarmingly high over the past 34 years, its ongoing pace has still considerably harmed forest health and environmental conditions. Moreover, there’s a noticeable upward trend in this degradation. The distribution of LST is closely associated with the distribution of NDVI, suggesting a strong inverse correlation between LST and NDVI for the years 1988, 1996, and 2009, with an upward trajectory. Jaafar et al. (2020) reported analogous findings in their research conducted in Perak and Kedah, Malaysia. They also uncovered a robust correlation between LULC and LST. In this study, it was noted that as the NDVI values increase, the LST decreases. Ranagalage, Estoque & Murayama (2017) and Dissanayake et al. (2019) have both demonstrated that the expansion of built-up areas and barren lands contributes to an elevation in LST. Our research findings suggest a notable impact on ECI when LST rises. This is evident as the majority of areas exhibiting high LST also correspond to areas with high ECI in the study area. The regression analysis unveiled a consistently robust positive correlation between ECI and LST across all four time points, as evidenced by the R2 values ranging from 0.8607 to 0.9640 (Fig. S4). In contrast to this pattern, the results revealed a significant inverse relationship between ECI and NDVI, with reported R2 values ranging from 0.4029 to 0.8421.

Recommendation and future research direction

Deforestation emerges as a critical issue throughout the Mannar district, as evidenced by previous research conducted by Rajendran (2019), Luxmini et al. (2020), Ranagalage et al. (2020), and Fernando & Edirisuriya (2016). However, our study was confined to the Musali DSD due to constraints in resources and time. However, it is imperative to monitor alterations in forest cover and vegetation quality in other areas in the district, like Madu DSD, employing high-resolution remote sensing data. Since most dry zone districts like Vavuniya, Mullaitivu, Jaffna and Kilinochchi were resettled following the civil war after 2009, studies on LULC change and deforestation analysis become increasingly important for making decisions related to the conservation of dry evergreen forests in those regions. This is attributed to a notable surge in resettlement and infrastructure development in those areas rapidly. Additionally, it is advisable to conduct a comparative study of various classification methods, such as support vector machine, k-mean algorithm, random forest, LandTrendr change detection algorithm, and neural networks, as demonstrated by Dissanayake et al. (2019), Rathnayake, Jones & Soto-Berelov (2020) in future research endeavors. This could enhance accuracy and aid in selecting the most reliable method for analysis. Although our study did not exclude water bodies and vegetation areas in ECI calculation, it is recommended to exclude them to obtain more reliable results, following similar approaches as those adopted in studies by Senanayake, Welivitiya & Nadeeka (2013), Ranagalage, Estoque & Murayama (2017), and Saputra, Jamadi & Sari (2023). VCI, is a widely utilized drought index for assessing vegetation drought stress. However, it may not be ideal for evaluating long-term trends in drought impacts on vegetation as emphasized by Yin et al. (2024) since VCI can inadvertently inherit the greening trend observed in NDVI. Therefore, it is advisable to conduct comparison studies on multi-source indicators of vegetation greenness and drought. Specifically, focusing on indices like Leaf Area Index (LAI), Palmer Drought Severity Index (PDSI), and Standardized Precipitation-Evapotranspiration Index (SPEI) as proposed by Yin et al. (2024). Given the inherent limitations of Landsat data in NDVI calculation, it is advisable to explore alternative methods and datasets, such as deep learning-based approaches and Synthetic Aperture Radar (SAR) data as highlighted by Malik, Shukla & Mishra (2019).

Conclusions

In our study, we endeavored to assess the dynamics in forest cover, vegetation health, and environmental conditions in Musali DSD utilizing Landsat time series data. The results obtained through supervised classification reveal a relatively moderate pace of deforestation spanning 34 years, with a total forest cover loss of 8.52 km2 primarily linked to the expansion of settlements and paddy fields. The conversion had a significant impact on the health of the vegetation in the area, as indicated by a notable decline in NDVI and a subsequent rise in LST. Vegetation stress has escalated due to the increasing environmental vulnerability, particularly in deforested areas over the past two decades. The findings suggest a noteworthy impact on ECI growth across the four time periods due to the rise in LST. The regression analysis revealed that the degradation of vegetation conditions contributed to the increase in ECI. Given time and resource limitations, the study exclusively concentrated on Musali DSD. The reliability of our LULC findings is influenced by the inherent drawbacks of the MLC algorithm, such as its vulnerability to the distribution of categories in feature space and the sampling selection. Additionally, calculating NDVI using Landsat data encounters challenges arising from atmospheric conditions, sunlight and cloud cover. These factors limit the efficacy of multispectral bands in accurately capturing land characteristics. Moreover, NDVI is constrained by its capacity to solely capture linear relationships between NIR and red spectral bands, thereby limiting its ability to account for higher-order relationships between spectral channels. Because, VCI might unintentionally incorporate the prolonged greening pattern detected in NDVI, it could potentially compromise the reliability of our results.

Supplemental Information

Supplemental Information 1 The relevant data used to run regression analysis of NDVI, LST, ECI

Supplemental Information 2 LULC information: (A) Forest cover (FC); (B) Agriculture (AG); (C) Built-up/settlements (BS); (D) Water bodies (WB); (E) Other

Supplemental Information 3 Land use conversion in Musali DSD: (A) 1988–1996; (B) 1996–2009; (C) 2009–2022; (D) 1988–2022

Supplemental Information 4 (A) new paddy fields on clear-cut area in Karadikkuli; (B) clear-cut area for plantation in Kondachchi; (C) clear-cut areas for settlements in Silawatura; (D) Kondachchi plantation land

Supplemental Information 5 Correlation between LST,NDVI and ECI,1988,1996,2009,2022

Supplemental Information 6 (A) New paddy fields on clear-cut area in Karadikkuli; (B) clear-cut area for plantation in Kondachchi; (C) clear-cut areas for settlements in Silawatura; (D) Kondachchi plantation land

Supplemental Information 7 Correlation between LST,NDVI and ECI,1988,1996,2009,2022

Supplemental Information 8 Specifications of Landsat data used for the study

Supplemental Information 9 LULC classes with codes and descriptions

*Image reference with respect to Fig. S1

Supplemental Information 10 Statistics of accuracy assessments of LULC classes

FC, Forest cover; AG, Agriculture; BS, Built-up/settlements; WB, Water bodies; OT, Others

Supplemental Information 11 Descriptive statistics of NDVI changes in Musali DSD

Supplemental Information 12 Descriptive statistics of retrieved LST changes in Musali DSD(°C)

The authors express gratitude to the anonymous reviewers and editors for their valuable comments provided on improving the quality of the manuscript. The authors are grateful to the U.S. Geological Survey (USGS) for providing open source Landsat data relevant to the study.

Additional Information and Declarations

Competing Interests

Author Contributions

Data Availability

The authors declare there are no competing interests.

Neel Chaminda Withanage conceived and designed the experiments, performed the experiments, analyzed the data, prepared figures and/or tables, authored or reviewed drafts of the article, and approved the final draft.

Prabuddh Kumar Mishra conceived and designed the experiments, analyzed the data, prepared figures and/or tables, and approved the final draft.

Kamal Abdelrahman performed the experiments, authored or reviewed drafts of the article, and approved the final draft.

Rajender Singh performed the experiments, authored or reviewed drafts of the article, and approved the final draft.

The following information was supplied regarding data availability:

The raw data and additional tables and figures are available in the Supplemental File.

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
