# Peer review of "Monitoring deforestation, forest health, and environmental criticality in a protected area periphery using Geospatial Techniques"

_PeerJ, doi:10.7717/peerj.17714_

## Round 0.1 · original submission · Major Revisions

Dear Authors,

Please read the reviewers comments carefully and address them fully in a revised manuscript.

Reviewer 1 ·

Basic reporting

Dear editor,
I have completed my assessment of the manuscript titled "Monitoring Deforestation, Forest Health, and Environmental Criticality in a Protected Area Periphery Using Geospatial Techniques" submitted to PeerJ. It represents a significant contribution to comprehending the diminishing tropical dryland forest landscape in Sri Lanka, elucidating its ramifications for protected area management and conservation efforts. It certainly merits publication.
The article concentrates on analyzing spatial-temporal variations in forest cover, with particular emphasis on forest health and environmental conditions in a designated site in northern Sri Lanka over a span of 34 years. This analysis is conducted by integrating NDVI (Normalized Difference Vegetation Index), LST (Land Surface Temperature), VCI (Vegetation Condition Index), and ECI (Environmental Condition Index) data derived from Landsat imagery. The manuscript is well written and expressed. It may be consider for publication after revision
Here, I summarize my comments as follows:
1. Abstract
a. The main objective and justification of the study have been well articulated and refined.

2. Introduction
a. The study has provided a clearer articulation of how it addresses existing research gaps and advances knowledge of remote sensing-based LULC change and deforestation-related studies. But, please address the following;
2.1 Reduce the word limit of the introduction section as much as possible to suffice for the intended readership.
2.2 Additionally, it may be beneficial to clearly articulate how the study addresses existing gaps in research.

3. Method:
a. The manuscript provides a good level of detail regarding the methods used for RS-based LULC/vegetation analysis. The methodology has provided step-by-step instructions or equations for various remote sensing-based indexes, enhancing replicability.
3.1 Please provide information on the software tools or platforms used for RS and statistical analysis that would also aid in reproducibility.

4. Results
a. Overall, by integrating well-presented graphs and figures into the manuscript, researchers could effectively communicate their findings.
b. The interpretation of results, discussion, and conclusion appears well-supported by the data presented in the manuscript.
5. Discussion
a. The study has provided additional context or implications of the findings within the broader context of forest and natural resource management in the human-dominated landscape.
5.1 But, please provide some additional information on future research direction supporting more similar previous research.

6. Conclusion
6.1 In the conclusion section, it is imperative to acknowledge the study's limitations, explicitly addressing constraints related to data availability or quality that could impact the findings. This includes acknowledging any potential limitations in the accuracy or resolution of the remote sensing data used.

7. Manuscript Structure and Flow:
a. Maps are well drawn with essential map elements. Graphs and tables match the manuscript details.
b. English language and formatting are ok.
c. Although the general framework of the manuscript appears satisfactory, it is recommended to include a section addressing limitations and outlining directions for future studies in the conclusion section.

8. Overall recommendations
These suggestions aim to enhance the clarity, replicability, and overall quality of the manuscript, thereby strengthening its contribution to the field of remote sensing-based land use and land cover change and forest condition assessment

Experimental design

No

Validity of the findings

No

Additional comments

The manuscript can be accepted for publication after revision.

Annotated reviews are not available for download in order to protect the identity of reviewers who chose to remain anonymous.

Reviewer 2 ·

Basic reporting

The article is interesting. However, it has numerous serious defects that must be addressed before the work can be published. See the "Additional comments". The design of some figures is poor, some are not well preparedand are not very visual. I have indicated to the authors which ones they should explicitly improve.

The quality of English must be significantly improved for the article to be considered for acceptance. There are inconsistencies in verb tenses. The punctuation of the text needs to be carefully reviewed. Some parts of the article (such as the final section of the abstract) are very difficult to understand due to the lack of punctuation marks. It is highly recommended that the work be thoroughly reviewed by a native speaker.

In general, the work is referenced in a way that is not bad, but could be improved. I indicate in the additional comments to the authors where references are missing, from my point of view.

The authors do not provide the basic data used for the study. Perhaps, it should be indicated in a specific section of "Data availability" how the images that have been the subject of analysis can be obtained.

Experimental design

The methods used seem to be sufficiently rigorous.

However, the materials and methods section has serious deficiencies that must be addressed. It is recommended to see the "Additional comments for authors" section.

Validity of the findings

The data used in the study is of sufficient quality.

There are some points that leave me somewhat confused, especially in the VCI calculation, and in the LST vs NDVI relationship for the 2022 image. In the additional comments I tell you what I think should be reviewed in relation to this.

The conclusions are supported by evidence.

Additional comments

Major points:

The quality of English must be significantly improved for the article to be considered for acceptance. There are inconsistencies in verb tenses. The punctuation of the text needs to be carefully reviewed. Some parts of the article (such as the final section of the abstract) are very difficult to understand due to the lack of punctuation marks. It is highly recommended that the work be thoroughly reviewed by a native speaker.

Also, the format of all mathematical expressions (superscripts, subscripts, etc.) should be thoroughly reviewed.

Introduction

1. From my point of view, I believe there should be a better introduction to each vegetation index used in the study (for example, explaining that NDVI is an index that relates reflectance in the infrared and red bands), so that readers not-specialized in the determination and measurement of these indexes can better understand what is being performed in this study. So please, explain more what are NDVI; VCI, LST and ECI. It would also be beneficial to indicate some methods of measurement. I leave you some references for your consideration: https://doi.org/10.1016/j.compag.2019.04.021, https://doi.org/10.3390/agriculture13081467, https://doi.org/10.3390/s18072083

2. Line 80-83: “Apart from those indices LST and NDVI are used to develop other indexes like vegetation condition index (VCI) to describe the vegetation quality.” Please, explain how, and reference. Do the same for: “Changing LST and NDVI is also an indicator of worsening environmental quality as well.”

3. Line 96-102: Please, consider rewriting the final part of introduction. It is assumed that the reader knows how to write a scientific article and how it should be organized. At the end of the introduction, you should indicate what your article is about and what is novel about it.

Materials and Methods – Study Area

1. This section is poorly written and needs to be almost entirely rewritten. Some of the information provided is irrelevant to the study. It's essential to filter out the information that is truly relevant to the study and eliminate the rest. Furthermore, the information presented lacks proper citation support. Therefore, a thorough revision is necessary to ensure that the work meets the minimum formal requirements for publication consideration.

2. Fig 1: "Where did you get this figure from? Does it have copyright? If you did not create it yourselves, you should reference it."

Materials and Methods – Materials

1. Which bands were used for each of the indexes determination? Please, specify.

Materials and Methods – Methods

1. To guarantee reproducibility, more details of the image pre-processing should be provided. It is not necessary to put it on the main text of the article. It could be added to the supplemenaty information.

2. Equation (1) sould be explained, and, at least, it should be indicated what each one of the variables represent.

3. Lines 165-169: This paragraph is difficult to understand. Please rewrite in a more comprehensible way.


Materials and Methods – Accuracy assesment

1. Define GCP.

2. Explain better kappa coefficient, user accuracy and producer accuracy, and better reference this part of the work. Ttry to make this part more comprehensible.

Materials and Methods – Deriving the Land Surface Temperature

1. Define DN.

2. If you use expressions that have been previously used by other authors, please provide references.

3. Please revise line 195, I think there are some mistakes.

Materials and Methods – NDVI

1. Please provide a some more references linking vegetation level or chlorophyll activity with NDVI.

2. Lines 216 – 218: I believe this classification should be revised. As far as I know, values between 0 and 0.3 can represent dying vegetation, or an área with only some vegetation. I don't think it's appropriate to solely classify NDVI as positive or negative.

3. Line 217: I think you mean “<-1 and >1 values were recorded as NODATA... In any case, you should never have obtained this result, since it it mathematically imposible...
Materials and Methods – VCI

1. If you take VCI equation from other authors, please reference.

Results

1. In order to assess whether the variations are above the error associated with the analysis, an evaluation of uncertainties should be performed, and each resulting value (e.g., forest area) should be provided with its error.

2. The Fig. S2 could go in the main body of the article. However, the figure is underdeveloped and not visually appealing. Perhaps, a pie chart for each year might better visualize the data.

3. In all tables, be consistent with significant figures.

4. Fig. S3 is not visual. Try to improve it. There is a loti f data and it is difficult to interpret anything.

5. Line 251: Search a reference for “specifically for paddy fields).

6. I believe the analysis part concerning NDVI should be reframed. I don't think it's particularly relevant to focus on the maximum and minimum NDVI values. For example, I think the evolution of the index, for instance, in the area previously identified as forest could be analysed. It doesn't make sense to discuss NDVI, for example, in urban areas. The NDVI value in the forest can provide information, for example, about vegetation density or the intensity of photosynthetic activity. In cities, it makes no sense to calculate the index.

7. Fig, 5 (NDVI): I believe that in order to better observe the changes in NDVI over time in each of the zones, the color scale of the four figures should have the same maximum and minimum values.

8. Fig, 5 (LST): I also suggest that the color scale of the four figures should have the same maximum and minimum values.

9. Fig. 6 The design of Fig. 6 is very poor and cannot be accepted. With such a large number of points, they should not be so thick as it's not possible to appreciate each one. The size of the points should be reduced. Please improve the aesthetics of the graph (axes, units, etc.).

10. Why do LST vs NDVI plot for 2022 present highest dispersión? It seems such as if there are two populations of points. Some of them on the direction expected, and others, at lower LST values...

11. In order to calculate VCI, what values are you taking as NDVI max and NDVImin, in each case? Please indicate the numerical values in your response to my comments (you do not need to include them in the article).

Discussion

1. Please evaluate whether the “Limitations and Future Research” section should remain as a discussion, or if in fact part of it should be incorporated into the conclusions

Minor points:

General: For all indexes, only write the full name once. Aftwerawards, use always only the abbreviation.
Line 51-54: Please rewrite this sentence. It is too long and not grammatically correct. For instance, I suggest revising it to: [...] This has led to the disappearance of 420 million hectares of forest between 1990 and 2020.
Line 54-56: This phrase is not clear either, from my point of view.
Line 68-70 and 70-73: This phrases are not clear, and lack of punctuation marks from my point of view.
Line 77: Please, use a reference to support the following: “Additionally, it can be used as a proxy for biomass accumulation”
Line 105: Please, indicate that DS is the abbreviation for divisional secretariat.
Line 137-149: This part is unnecessarily long, and need to be summarized
Fig S1. Longitud misses º. Correct mark for degrees is º, not 0
Fig 2. Resolution must be clearly improved
Line 184: Please improve the grammar of this sentence.
Line 205: Is [8,17] a reference that you intended to include?
Be consistent with the referencing style throughout the entire manuscript

---

## Round 0.2 · Minor Revisions

The authors have addressed reviewers comments well, and the revised MS is substantially improved. However, I am returning it to the authors for one minor revision. On line 224 you state
" Where m=(Ɛ−Ɛ)−(1−Ɛσ)FƐv"
(Ɛ−Ɛ) is always zero; I suspect a subscript is missing. If you will address this quickly with a revised MS and a quick explanation to the editor, I believe the article will then be ready to accept.

---

## Round 0.3 · accepted · Accept

Thanks for your patience.